

# *Paenibacillus* strains with nitrogen fixation and multiple beneficial properties for promoting plant growth

Xiaomeng Liu, Qin Li, Yongbin Li, Guohua Guan and Sanfeng Chen

State Key Laboratory for Agrobiotechnology and College of Biological Sciences, China Agricultural University, Beijing, China

## ABSTRACT

*Paenibacillus* is a large genus of Gram-positive, facultative anaerobic, endospore-forming bacteria. The genus *Paenibacillus* currently comprises more than 150 named species, approximately 20 of which have nitrogen-fixation ability. The $N_2$-fixing *Paenibacillus* strains have potential uses as a bacterial fertilizer in agriculture. In this study, 179 bacterial strains were isolated by using nitrogen-free medium after heating at 85 °C for 10 min from 69 soil samples collected from different plant rhizospheres in different areas. Of the 179 bacterial strains, 25 *Paenibacillus* strains had *nifH* gene encoding Fe protein of nitrogenase and showed nitrogenase activities. Of the 25 $N_2$-fixing *Paenibacillus* strains, 22 strains produced indole-3-acetic acid (IAA). 21 strains out of the 25 $N_2$-fixing *Paenibacillus* strains inhibited at least one of the 6 plant pathogens *Rhizoctonia cerealis*, *Fusarium graminearum*, *Gibberella zeae*, *Fusarium solani*, *Colletotrichum gossypii* and *Alternaria longipes*. 18 strains inhibited 5 plant pathogens and *Paenibacillus* sp. SZ-13b could inhibit the growth of all of the 6 plant pathogens. According to the nitrogenase activities, antibacterial capacities and IAA production, we chose eight strains to inoculate wheat, cucumber and tomato. Our results showed that the 5 strains *Paenibacillus* sp. JS-4, *Paenibacillus* sp. SZ-10, *Paenibacillus* sp. SZ-14, *Paenibacillus* sp. BJ-4 and *Paenibacillus* sp. SZ-15 significantly promoted plant growth and enhanced the dry weight of plants. Hence, the five strains have the greater potential to be used as good candidates for biofertilizer to facilitate sustainable development of agriculture.

## INTRODUCTION

Nitrogen is an essential element to affect the yields of crops by influencing leaf area development and photosynthetic efficiency (*Fang et al., 2018*). The application of chemical nitrogen fertilizer can improve soil fertility and thus agricultural production. High rates of nitrogen fertilizer might boost yields, but can reduce the quality of agricultural products. However, approximately 100 Tg chemical nitrogen is applied in agricultural products every year, while only 17 Tg nitrogen is accounted for in crops (*Erisman et al., 2008*). Excessive use of chemical fertilizer has resulted in seriously negative impacts, such as soil

Corresponding author
Sanfeng Chen, chensf@cau.edu.cn

hardening and acidification, increased greenhouse gas (N$_2$O) emissions and enhanced nitrogen deposition (*Jiao et al., 2018*; *Reay et al., 2012*).

The *Paenibacillus* genus was first reclassified as a separate genus on the basis of the 16S rRNA gene sequences by *Ash, Priest & Collins (1993)*. Since its creation, the *Paenibacillus* genus embody more than 100 validly named species. Approximately 20 members of the *Paenibacillus* genus had been reported to have the capacity of fixing nitrogen, such as: *Paenibacillus polymyxa*, *Paenibacillus macerans*, *Paenibacillus azotofixans*, *Paenibacillus sabinae*, *Paenibacillus sonchi*, *Paenibacillus forsythia*, *Paenibacillus sophorae*, *Paenibacillus taohuashanense* and *Paenibacillus beijingensis* (*Grau & Wilson, 1962*; *Hong et al., 2009*; *Jin, Lv & Chen, 2011*; *Ma et al., 2007*; *Ma & Chen, 2008*; *Seldin, Van Elsas & Penido, 1984*; *Wang et al., 2014*; *Witz, Detroy & Wilson, 1967*; *Xie et al., 2012*). *Paenibacillus* is a group of Gram-positive, aerobic or facultative anaerobic, rod-shaped, endospore-forming bacteria. The widely distributed *Paenibacillus* bacteria could tolerate extreme environments and interact with a variety of plants (*Navarro-noya et al., 2012*). Currently, some *Paenibacillus* strains play a great role in agriculture and industry (*Seldin, 2011*).

Plant rhizosphere is a habitat of functional microorganisms, which encompasses a complex and dynamic zone of interactions between networks of organisms and their plant hosts (*Garcia & Kao-Kniffin, 2018*; *Zhalnina et al., 2018*). A large amount of strains isolated from plant rhizospheres are able to directly or indirectly promote plant growth, development and evolution, which are termed as plant growth-promoting rhizobacteria (PGPR) (*Mohamed et al., 2019*). PGPR can stimulate plant growth by a diversity of mechanisms including fixing nitrogen from atmosphere, solubilizing phosphorus, synthesizing siderophore, producing antimicrobial substances (antibiotics, bacteriocins and small peptides) and plant hormones such as indole, cytokinins or gibberellins (*Graham et al., 2000*; *Neilands, 1993*). Given these advantages, PGPR are widely used in sustainable agriculture to promote plant growth and control fungal pathogens (*Verma et al., 2018*). Some of *Paenibacillus* species can influence plant growth by one or more of mechanisms mentioned above (*Li et al., 2017*; *Weselowski et al., 2016*; *Xie et al., 2016*). Nowadays, with the rapid growth of population, most regions have increased the cereals production by the overuse of fertilizers, which not only accounts for a larger percentage of farmers' expenses but also increase risks of negative effect on environment (*Curatti & Rubio, 2014*; *Ivleva et al., 2016*; *Tayefeh et al., 2018*). It is the best choice to select the environmentally friendly *Paenibacillus* strains to substitute for chemical fertilizer due to its broad host range and its ability to secrete plant growth-enhancing substances and produce different kinds of antimicrobial substances (*Cho et al., 2007*; *Da Mota, Gomes & Seldin, 2008*; *Fortes et al., 2008*; *Li et al., 2007*; *Timmusk et al., 2009*).

The *Paenibacillus* strains have the potential to increase agricultural productivity, including weight of crops and root growth. The main purpose of this research was to isolate and identify *Paenibacillus* strains, to study the effect of these isolates on plant growth, and then to select the potential bacterial strains to be used in sustainable development of agricultural production.

## MATERIALS & METHODS

### Sample collection, isolation procedures and culture conditions

Sixty-nine soil samples were collected from various plant rhizospheres in different areas of China, which were described in Table 1 in detail. The soil samples were diluted gradiently by 0.9% saline solution (up to $10^{-5}$) and then screened on nitrogen-free medium after heating at 85 °C for 10 min. Three replicates per dilution were made. The nitrogen-free medium contained 20 g sucrose, 0.1 g $K_2HPO_4$, 0.4 g $KH_2PO_4$, 0.2 g $MgSO_4$ $7H_2O$, 0.01 g NaCl, 0.01 g $FeCl_3$, 0.002 g $Na_2MoO_4$ and 1.2–1.4 g agar per litre of water. Single colony for each possible species was selected after cultivation for 3–5 days at 30 °C. To reduce the influence of nitrogen from the soils and purify the strains, the isolates were transferred to the fresh nitrogen-free medium. The strains isolated in this study and their sources were listed in Table 1. All isolates are stored in our lab, and 16S rRNA sequences are available in database of GenBank.

### Amplification, cloning and sequencing of *nifH* gene

PCR amplification of *nifH* gene was carried out using the following primers: forward 5′-GGCTGCGATCC(CGA)AAGGCCGATC(CGA)ACCCG-3′ and reverse 5′-CTG(GCA)GCCTTGTTTCGCGGAT(CG)GGCATGGC-3′ as described by *Ding et al., (2005)*. The *nifH* gene fragments were purified using TIANgel Midi Purification Kit (Tiangen Biotech Co., LTD. Cat. #DP210, Beijing, China) and ligated to vector pGEM-T (Promega Co., Cat. #R6881; Madison, WI, USA) at 16 °C overnight. Recombinant plasmids were transformed into *Escherichia coli* JM109 and transformants were selected by blue/white screening procedure. Plasmids containing *nifH* gene were extracted and purified. Purified plasmids were then sequenced using the M13F and M13R primers by Shanghai Majorbio Bio-pharm Technology Co., LTD, Shanghai, China.

### Morphological characterization of strains

For observation of colony morphology, the bacterial strains were spread on Luria-Bertani (LB) agar. After incubation at 37 °C overnight, single colony was observed. Cell morphology was viewed by optical microscopy (Olympus, CX22LED, Tokyo, Japan).

### Sequence analysis and construction of the phylogenetic trees

All strains were cultured in LB broth medium overnight. After collection of bacteria by centrifugation, genomic DNA of isolates was extracted and purified using the TIANamp Bacteria DNA Kit (Tiangen Biotech Co., LTD. Cat. #DP302) according to the manufacturer's instructions. The amplication of 16S rRNA genes was performed with the universal primers: 27F (5′-AGAGTTTGATC(AC)TGGCTCAG-3′) and 1492R (5′-CGG(CT)TACCTTGTTACGACTT-3′) as described by *Khan et al. (2014)*. Then the 16S rRNA gene fragments were ligated into vector pGEM-T (Promega Co., Cat. #R6881) and sequenced by Shanghai Majorbio Bio-pharm Technology Co., LTD. The sequences of 16S rRNA gene were submitted to nucleotide database of GenBank and the accession numbers were displayed in Table 1. And the sequences were aligned with BLAST software from NCBI (https://blast.ncbi.nlm.nih.gov/Blast.cgi).

Liu et al. (2019), *PeerJ*, DOI 10.7717/peerj.7445

**Table 1** Characterization and nitrogenase activity of isolates.

| Isolates | Cell morphology | Colony morphology | Nitrogenase activity[a] | GenBank accession number | Origin and location |
|---|---|---|---|---|---|
| *Paenibacillus* sp. BJ-2 | Rods | Moist, milky | $1{,}085.61 \pm 75.64^{ghi}$ | MF967282 | Jujube, mountain in Huairou, Beijing 40°32′N, 116°62′E |
| *Paenibacillus* sp. SZ-1a | Rods | Moist, milky | $118.65 \pm 3.97^{k}$ | MF967283 | Maize, farmland in Changping, Beijing 40°22′N, 116°20′E |
| *Paenibacillus* sp. SZ-1b | Rods | Moist, milky | $1{,}1868.65 \pm 1740.55^{a}$ | MF967284 | Maize, farmland in Changping, Beijing 40°22′N, 116°20′E |
| *Paenibacillus* sp. BJ-4 | Rods | Dry, milky | $1{,}296.94 \pm 439.17^{g}$ | MF967285 | Apple, orchard in Shunyi, Beijing 40°13′N, 116°65′E |
| *Paenibacillus* sp. BJ-5 | Rods | Dry, white | $468.63 \pm 42.20^{hijk}$ | MF967286 | Persimmon, mountain in Shunyi, Beijing 40°13′N, 116°65′E |
| *Paenibacillus* sp. SZ-8 | Rods | Moist, milky | $1{,}131.54 \pm 15.92^{gh}$ | MF967287 | Maize, field in Changping, Beijing 40°22′N, 116°20′E |
| *Paenibacillus* sp. BJ-7 | Rods | Moist, milky | $314.60 \pm 19.18^{jk}$ | MF967288 | Wheat, farmland in Miyun, Beijing 40°37′N, 116°85′E |
| *Paenibacillus* sp. SZ-10 | Short rods | Moist, milky | $371.28 \pm 7.67^{ijk}$ | MF967289 | Maize, farmland in Changping, Beijing 40°22′N, 116°20′E |
| *Paenibacillus* sp. SZ-11 | Rods | Moist, milky | $857.47 \pm 114.89^{ghij}$ | MF967290 | Pepper, herbary in Changping, Beijing 40°22′N, 116°20′E |
| *Paenibacillus* sp. SZ-13a | Rods | Dry, milky | $9{,}731.36 \pm 259.71^{b}$ | MF967291 | Medicinal plant, farmland in Changping, Beijing 40°22′N, 116°20′E |
| *Paenibacillus* sp. SZ-13b | Rods | Dry, milky | $3{,}131.89 \pm 100.61^{e}$ | MF967292 | Medicinal plant, farmland in Changping, Beijing 40°22′N, 116°20′E |
| *Paenibacillus* sp. SZ-15 | Rods | Moist, milky | $1{,}316.19 \pm 36.64^{g}$ | MF967293 | Wheat, farmland in Changping, Beijing 40°22′N, 116°20′E |
| *Paenibacillus* sp. SZ-16 | Rods | Moist, milky | $444.73 \pm 119.11^{hijk}$ | MF967294 | Spinach, herbary in Changping, Beijing 40°22′N, 116°20′E |
| *Paenibacillus* sp. BJ-6 | Short rods | Dry, milky | $176.7 \pm 29.43^{jk}$ | MF967295 | Bamboo, mountain in Huairou, Beijing 40°32′N, 116°62′E |
| *Paenibacillus* sp. AH-1 | Short rods | Moist, milky | $192.43 \pm 73.08^{jk}$ | MF967296 | Grape, orchard in Hefei, Anhui 31°86′N, 117°27′E |
| *Paenibacillus* sp. SZ-14 | Rods | Moist, milky | $331.95 \pm 22.73^{jk}$ | MF967297 | Rice, farmland in Changping, Beijing 40°22′N, 116°20′E |
| *Paenibacillus* sp. YN-3 | Short rods | Moist, white | $3{,}201.92 \pm 104.96^{e}$ | MF967298 | Sugarcane, farmland in Pu'er, Yunnan 23°07′N, 110°03′E |
| *Paenibacillus* sp. AH-3 | Short rods | Moist, white | $57.23 \pm 14.44^{k}$ | MF967299 | Arbor, natural forest in Wuhu, Anhui 31°95′N, 118°73′E |
| *Paenibacillus* sp. AH-4 | Short rods | Moist, white | $6{,}514.37 \pm 997.12^{c}$ | MF967300 | Arbor, natural forest in Hefei, Anhui 31°95′N, 118°73′E |
| *Paenibacillus* sp. YB-3 | Rods | Moist, milky | $733.92 \pm 49.28^{ghijk}$ | MF967301 | Fruit, mountain in Yibin, Sichuan 28°77′N, 104°62′E |
| *Paenibacillus* sp. WF-6 | Rods | Moist, milky | $2{,}081.30 \pm 340.66^{f}$ | MF967302 | Wheat, field in Weifang, Shandong 36°62′N, 119°10′E |
| *Paenibacillus* sp. JS-4 | Rods | Moist, milky | $6{,}843.56 \pm 365.69^{c}$ | MF967303 | Reed, countryside in Suzhou, Jiangsu 31°32′N, 120°62′E |
| *Paenibacillus* sp. HN-1 | Short rods | Moist, milky | $4{,}476.80 \pm 306.64^{d}$ | MF967304 | Rice, farmland in Xiangtan, Hunan 27°52′N, 112°53′E |
| *Paenibacillus* sp. CD-4a | Rods | Moist, milky | $272.67 \pm 14.24^{jk}$ | MF967305 | Rape, field in Chengdu, Sichuan 30°67′N, 104°07′E |
| *Paenibacillus* sp. CD-4b | Short rods | Moist, milky | $5{,}174.69 \pm 478.7^{d}$ | MF967306 | Fruit, mountain in Chengdu, Sichuan 30°67′N, 104°07′E |

**Notes.**

[a]The unit of nitrogenase activity is nmol $C_2H_4$ mg$^{-1}$ protein h$^{-1}$.

Results are means $\pm$ SE of 3 independent biological replicates. Different letters are significantly different from each other according to the least significant differences (LSD) test ($P < 0.05$).

The phylogenetic tree was constructed from evolutionary distance matrices using the neighbor-joining method with MEGA6 software package (*Tamura et al., 2013*). Bootstrap analysis was performed with 1,000 cycles, and only bootstrap values greater than 50% were shown at the branch points.

### Nitrogenase activity assay

For determination of the nitrogenase activity, strains were grown in 20 mL of LB broth medium in 50 mL flasks shaken at 200 rpm overnight at 37 °C. The cultures were collected by centrifugation, precipitations were washed three times with sterilized water and then resuspended in nitrogen-limited medium (per liter: 26.3 g $Na_2HPO_4 12H_2O$, 3.4 g $KH_2PO_4$, 26 mg $CaCl_2 2H_2O$, 30 mg $MgSO_4$, 0.3 mg $MnSO_4$, 36 mg ferric citrate, 7.6 mg $Na_2MoO_4$ $2H_2O$, 10 $\mu$g *p*-aminobenzoic acid, 10 $\mu$g biotin, 0.4% (w/v) glucose and 0.03% (w/v) glutamic acid). The nitrogenase activity was determined using the acetylene reduction assay and expressed as nmol $C_2H_4$ mg$^{-1}$ protein h$^{-1}$ (*Wang et al., 2013*; *Wang et al., 2018*).

### Assessment of antagonistic activity against plant pathogens

The assessment of the *Paenibacillus* strains isolated from the rhizospheres for antagonism against 6 plant pathogens including *Rhizoctonia cerealis* (ACCC 37393), *Fusarium graminearum* (ACCC 36249), *Gibberella zeae* (CGMCC 3.2873), *Fusarium solani* (CGMCC 3.17848), *Colletotrichum gossypii* (CGMCC 3.1859) and *Alternaria longipes* (CGMCC 3.2875), was performed in agar plate assay using potato dextrose agar (PDA). The fungal pathogens were inoculated in the center of the agar plate, and the *Paenibacillus* strains were placed at a distance of 3.5 cm from the center of the plate. After 3–7 days of incubation at 30 °C, the plates were examined and measured for fungal pathogens growth inhibited zones around the *Paenibacillus* strains. All tests were carried out in three duplicates.

### Measurement of indole-3-acetic acid (IAA) production

The ability of producing IAA was assessed by colorimetric analysis. For the measurement of IAA production, the tested strains were grown in 20 mL King B broth medium (per liter: peptone, 20 g; $K_2HPO_4$, 1.15 g; $MgSO_4$ $7H_2O$, 1.5 g; glycerol, 10 g) supplemented with 100 $\mu$g mL$^{-1}$ Trp (IAA precursor). The non-cultured medium was used as the negative control and *Azospirillum brasilense* SP7 was selected as the positive control. The culture supernatants were obtained by centrifuging at 12,000 rpm for 10 min. The test strains were measured by colorimetric assay according to the method described by *Glickmann & Dessaux (1995)*. Briefly, two mL Salkowski reagent containing 4.5 g/L $FeCl_3$ in 10.8 M $H_2SO_4$ was mixed with one mL supernatant. Then, the mixture was stirred evenly and left in the darkness for 30 min at room temperature. The production of IAA was measured using spectrophotometer (Shimadzu UVmini-1240; Kyoto, Japan) at 530 nm. Each treatment had three biological replicates.

### Evaluation of plant growth-promoting effect

The tested strains were evaluated for their potential to promote plant growth on wheat cultivar Jimai 22 (Shandong Runfeng Seed Industry Co., Ltd., Shandong Sheng, China), cucumber Zhongnong 8 (Beijing Shengfeng Garden Agricultural Technology Co., Ltd.,

Beijing, China) and tomato Jiafen 15 (Tianjin Xingke Seed Co., Ltd., Tianjin, China) seedlings in the greenhouse of China Agricultural University, Beijing, China. The lengths and dry weights of three plants inoculated with strains were determined by the procedure described by *Li et al. (2017)*.

For preparing the bacterial cultures, each isolate was grown 150 mL LB broth medium for 24 h at 30 °C. After incubation, the cells were harvested by centrifugation at 6,000 rpm for 5 min at room temperature. The cell pellet was washed with sterile water and then adjusted to $10^8$ cells mL$^{-1}$ with 0.9% saline solution.

Wheat, cucumber and tomato seeds were sterilized with 10% sodium hypochlorite for 10 min and washed with sterilized water three times. Then the seeds germinated on sterile wet filter in Petri dishes in the dark at 25 °C for 5–7 days. After germination, seedlings were soaked in bacterial suspensions ($10^8$ cells mL$^{-1}$) for 15 min. Then three seedlings of different plants were transplanted into 12-cm-diam pots containing in the medium of turfy soil (Beijing Jixiang Feiyun Garden Engineering Co., Ltd. Cat. #101G): vermiculite (Beijing Jixiang Feiyun Garden Engineering Co., Ltd. Cat. #GM010108) of 1:1, and grown in the greenhouse (16 h day/8 h night and 22 °C/10 °C day/night temperature). Each treatment had three pots. Two weeks later, each of the seedlings was watered with 15 mL bacterial suspensions ($10^8$ cells mL$^{-1}$) again. The un-inoculated seedlings were used as negative controls, while the un-inoculated seedlings watered with nitrogen fertilizer (83 mg N kg$^{-1}$ soil) were set as positive controls (*Li et al., 2019*). After five-week growth, the plants were harvested and the roots were washed carefully with running water to remove the adherent soil. The lengths of the shoot and root and dry weights of the shoot and root were recorded and statistically analyzed, respectively.

### Statistical analysis

Each treatment had three replicates. Statistical analysis was performed using SPSS 20.0 (SPSS, Chicago, IL, USA). Means of different treatments were compared using the least significant difference (LSD) at 0.05 level of probability.

### Ethics approval and consent to participate

Not applicable.

## RESULTS

### The *nifH* gene analysis and nitrogenase activity assay

Nitrogenase is comprised of two component proteins: Fe protein and MoFe protein (*Mus et al., 2018*). The Fe protein is encoded by *nifH* gene, and MoFe protein is encoded by *nifD* and *nifK* genes. The conserved *nifH* gene has been exploited to screen the genetic potential for nitrogen-fixing bacteria in the environment (*Ding et al., 2005*; *Mehta, Butterfield & Baross, 2003*).

In this study, 179 strains were isolated by using nitrogen-free medium after heating at 85 °C for 10 min from 69 soil samples collected from different plant rhizospheres in different areas. PCR amplification of *nifH* gene (encoding Fe protein of nitrogenase) with universal primers was conducted using genomic DNA extracted from above bacteria. The

results showed that a *nifH* gene fragment of 323 nucleotides was detected in 25 isolates (Table 1). The PCR-amplified *nifH* gene fragments from 25 isolates were sequenced and their predicted amino acid sequences of NifH were aligned with the NifH sequences from other diazotrophs. The results showed that all of them except for *Paenibacillus* sp. HN-1 shared 84%–99% NifH sequence identity with other *Paenibacillus* strains. The sequencing result of *Paenibacillus* sp. HN-1 *nifH* fragment displayed double peaks, which indicated that there were multiple *nifH* genes in its genome.

As displayed in Table 1, all of the 25 strains with *nifH* genes had nitrogenase activities with variation from 57.23 to 11,868.65 nmol $C_2H_4$ $mg^{-1}$ protein $h^{-1}$. *Paenibacillus* sp. SZ-1b presented the highest nitrogenase activity (11868.65 nmol $C_2H_4$ $mg^{-1}$ protein $h^{-1}$). *Paenibacillus* sp. SZ-13a, *Paenibacillus* sp. SZ-13b, *Paenibacillus* sp. YN-3, *Paenibacillus* sp. AH-4, *Paenibacillus* sp. JS-4 and *Paenibacillus* sp. CD-4b had higher nitrogenase activities (>,3000 nmol $C_2H_4$ $mg^{-1}$ protein $h^{-1}$). The nitrogenase activity, cell morphology, colony morphology, GenBank accession number and origin/location were listed in Table 1.

## Sequencing and phylogeny of 16S rRNA

The 16S rRNA gene sequence is named as the evolution clock of bacterial phylogeny because of high conservation and slow evolution, which is widely used in identification of bacteria (*Roller et al., 1994*; *Vandamme et al., 1996*). The 16S rRNA gene sequences of the 25 strains were compared with the datebase reserved in GenBank (https://www.ncbi.nlm.nih.gov/genbank/). The alignment results indicated all of the isolates were *Paenibacillus*. The GenBank accession numbers of them after the bacterial names were shown in Table 1.

A phylogenetic tree was constructed based on 16S rRNA sequence, which branched into five clusters on the basis of the distance data. The cluster I totally including 17 isolates formed a larger cluster with *P. polymyxa*, *Paenibacillus jamilae* and *Paenibacillus peoriae*. Among the 17 isolates, six isolates exhibited 99.2%–99.6% 16S rRNA sequence similarities with *P. polymyxa*. 7 isolates had the highest similarities with *P. jamilae*, and four isolates showed particularly high homologies with *P. peoriae* (>99.5%). The cluster II contained three isolates, which displayed the highest similarity with *Paenibacillus brasilensis*, ranging from 99% to 99.2%. The cluster III only included *Paenibacillus* sp. CD-4a, which had highest 16S rRNA sequence similarity with *Paenibacillus jilunlli* (99.6%). The cluster IV which consisted of two strains clustered with *Paenibacillus zanthoxyli* showing 99.3% to 99.6% 16S rRNA sequence similarities with *P. zanthoxyli*. The cluster V covering two isolates formed a monophyletic cluster with *Paenibacillus stellifer* bacteria, and their 16S rRNA sequences similarities with *P. stellifer* were above 99%.

## Antibacterial capacity determination

In the study, all 25 *Paenibacillus* strains were tested against six plant pathogens. The results (Table 2) showed that 21 bacteria presented antibiosis, inhibiting at least one of the 6 indicator phytopathogens. Out of them, 18 bacteria could inhibit five plant pathogens (*R. cerealis*, *F. graminearum*, *G. zeae*, *C. gossypii* and *A. longipes*). Furthermore, *Paenibacillus* sp. SZ-13b exhibited an extremely good antibiotic activity, which was able

**Table 2** Antimicrobial activity of the *Paenibacillus* strains, which inhibit 6 indicator bacteria.

| Strains | R. cer | F. gra | G. zeae | F. sol | C. gos | A. lon |
|---|---|---|---|---|---|---|
| *Paenibacillus* sp. BJ-2 | ++ | ++ | ++ | – | +++ | + |
| *Paenibacillus* sp. SZ-1a | ++ | ++ | ++ | – | ++ | ++ |
| *Paenibacillus* sp. SZ-1b | +++ | +++ | ++ | – | ++ | +++ |
| *Paenibacillus* sp. BJ-4 | ++ | +++ | ++ | – | ++ | +++ |
| *Paenibacillus* sp. BJ-5 | – | +++ | + | – | ++ | ++ |
| *Paenibacillus* sp. SZ-8 | ++ | +++ | ++ | – | +++ | ++ |
| *Paenibacillus* sp. BJ-7 | ++ | +++ | +++ | – | ++ | + |
| *Paenibacillus* sp. SZ-10 | ++ | +++ | +++ | – | ++ | ++ |
| *Paenibacillus* sp. SZ-11 | ++ | ++ | ++ | – | ++ | ++ |
| *Paenibacillus* sp. SZ-13a | +++ | ++ | ++ | – | ++ | ++ |
| *Paenibacillus* sp. SZ-13b | ++ | ++ | ++ | + | ++ | |
| *Paenibacillus* sp. SZ-15 | +++ | ++ | +++ | – | +++ | ++ |
| *Paenibacillus* sp. SZ-16 | ++ | ++ | ++ | – | ++ | ++ |
| *Paenibacillus* sp. BJ-6 | ++ | ++ | – | + | ++ | – |
| *Paenibacillus* sp. AH-1 | +++ | +++ | +++ | – | ++ | ++ |
| *Paenibacillus* sp. SZ-14 | ++ | +++ | ++ | – | +++ | ++ |
| *Paenibacillus* sp. YN-3 | ++ | ++ | ++ | – | ++ | ++ |
| *Paenibacillus* sp. AH-3 | – | – | – | – | – | – |
| *Paenibacillus* sp. AH-4 | + | – | – | – | – | – |
| *Paenibacillus* sp. YB-3 | ++ | +++ | ++ | – | ++ | +++ |
| *Paenibacillus* sp. WF-6 | ++ | ++ | ++ | – | ++ | ++ |
| *Paenibacillus* sp. JS-4 | ++ | +++ | ++ | – | +++ | +++ |
| *Paenibacillus* sp. HN-1 | – | – | – | – | – | – |
| *Paenibacillus* sp. CD-4a | – | – | – | – | – | – |
| *Paenibacillus* sp. CD-4b | – | – | – | – | – | – |

**Notes.**
R. cer, R. cerealis; F. gra, F. graminearum; G. zeae, G. zeae; F. sol, F. solani; C. gos, C. gossypii; A. lon, A. longipes; (–), no inhibition; (+), inhibition zone diameters from 5 to 15 mm; (++), inhibition zone diameters from 15 to 25 mm; (+++), inhibition zone diameters from 25 to 35 mm.

to inhibit the growth of all indicator phytopathogens. The growth of *F. graminearum* was strongly inhibited, showing the average inhibition zones larger than 25 mm. While the growth of *F. solani* was weakly inhibited, which was only inhibited by two strains (*Paenibacillus* sp. SZ-13b and *Paenibacillus* sp. BJ-6) with the inhibition zones around 5 and 15 mm. In addition, *Paenibacillus* sp. AH-3, *Paenibacillus* sp. HN-1, *Paenibacillus* sp. CD-4a and *Paenibacillus* sp. CD-4b could not exhibit any antibiotic effect on six indicator fungi.

In general, out of the 25 tested strains, 80% strains presented antimicrobial activity against plant pathogens, with average inhibition zones varying from 15 to 35 mm. Combination with their phylogeny of 16S rRNA, the isolates with inhibition flocked together, which were particularly close to *P. polymyxa* and its highly close species (Fig. 1).

## Assessment of IAA production and plant growth promoting traits

IAA is an essential plant hormone regulating the growth and development of plants. In this study, we determined the ability of producing IAA for all strains. Figure 2 showed

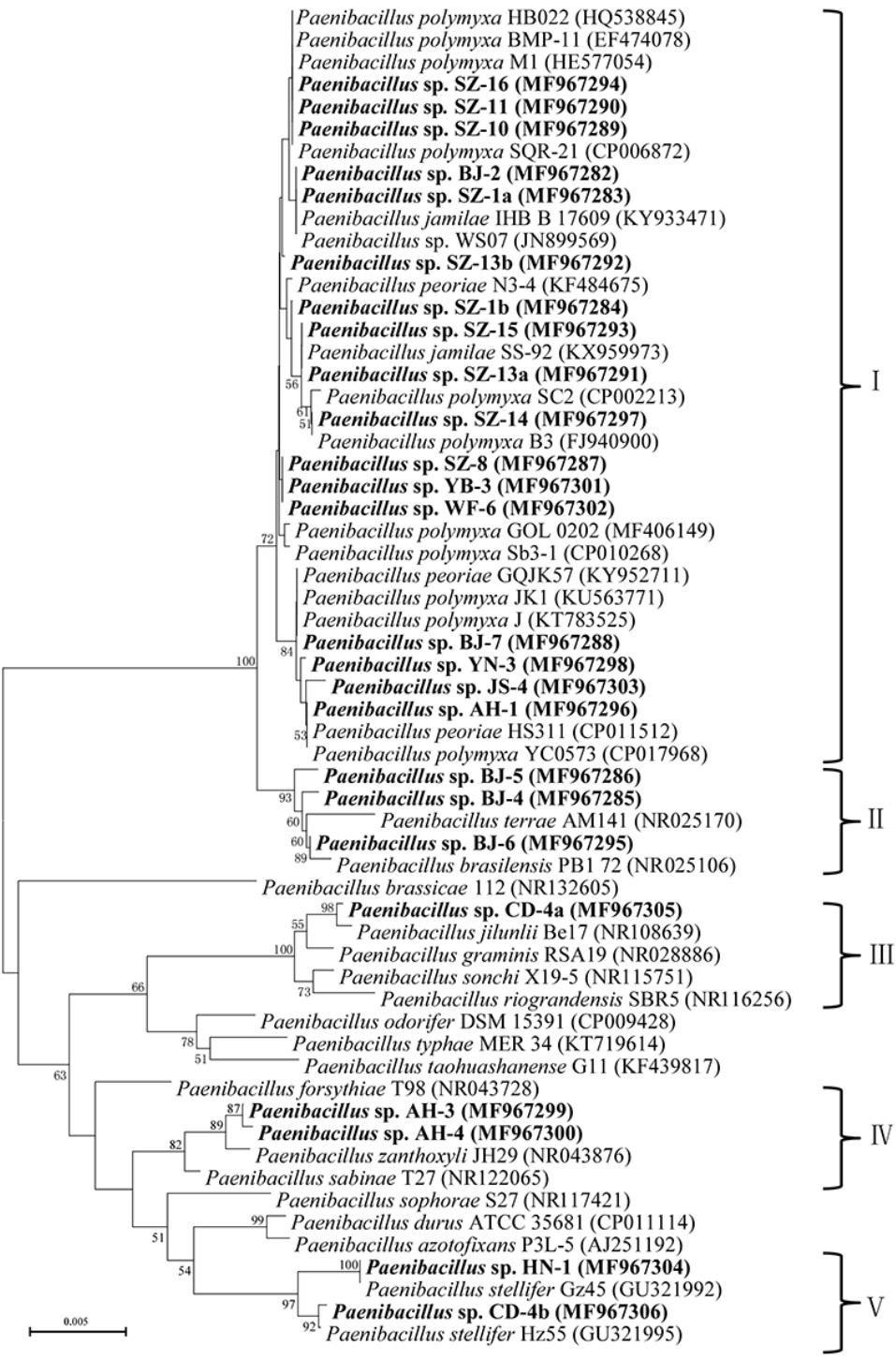

**Figure 1** **Neighbour-joining phylogenetic tree based on 16S rRNA sequence showing the position of isolated strains with other closely related strains of the genus _Paenibacillus_ in GenBank.** The tree was structured using neighbor joining method, with the bootstrap percentage values obtained from 1,000 cycles. Only bootstrap values greater than 50% are shown at the branching points. Bar, 0.005 substitutions per nucleotide position. Isolated strains in this study are underlined with the bold letters.

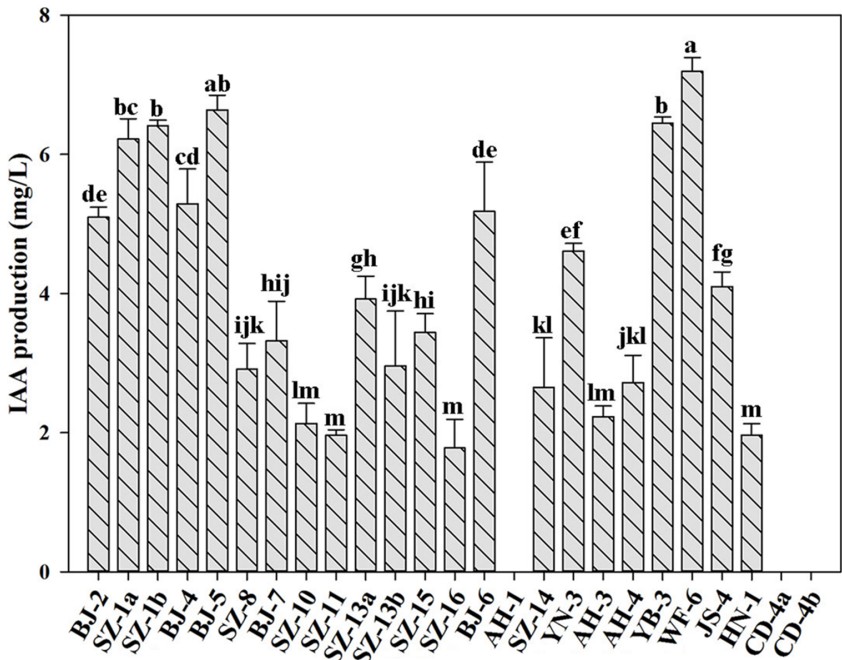

**Figure 2** **Qualitative analysis of IAA production by isolated strains.** Data are means ± SE of threein-dependent biological replicates. Bearing different alphabets are significantly different from each other according to the LSD test ($p < 0.05$).

that besides *Paenibacillus* sp. AH-1, *Paenibacillus* sp. CD-4a and *Paenibacillus* sp. CD-4b, the rest of tested strains were capable of producing IAA. Out of them, *Paenibacillus* sp. WF-6 produced the highest yield of IAA (7.19 mg L$^{-1}$). In addition, the other nine bacteria (*Paenibacillus* sp. BJ-2, *Paenibacillus* sp. SZ-1a, *Paenibacillus* sp. SZ-1b, *Paenibacillus* sp. BJ-4, *Paenibacillus* sp. BJ-5, *Paenibacillus* sp. BJ-6, *Paenibacillus* sp. YN-3, *Paenibacillus* sp. YB-3, *Paenibacillus* sp. JS-4) could yield relatively high amount of IAA (>4 mg L$^{-1}$).

According to above results of nitrogenase activities, antibacterial capacities and IAA production, we chose eight strains (*Paenibacillus* sp. SZ-1b, *Paenibacillus* sp. BJ-4, *Paenibacillus* sp. SZ-10, *Paenibacillus* sp. SZ-13b, *Paenibacillus* sp. SZ-14, *Paenibacillus* sp. YB-3, *Paenibacillus* sp. WF-6, *Paenibacillus* sp. JS-4) to assess their capabilities of promoting growth of plants (wheat, cucumber and tomato). Inoculation of plants with some *Paenibacillus* isolates appeared to promote plant growth including plant height and dry weight (Figs. 3 and 4). As shown in Fig. 4A, wheat seedlings inoculated with *Paenibacillus* sp. JS-4 led to a maximum increase (30.9%) in shoot length, followed by *Paenibacillus* sp. SZ-1b (23.3%) and *Paenibacillus* sp. BJ-4 (22.3%). While inoculation with *Paenibacillus* sp. SZ-14 yielded a maximum increase (54.2%) in root length, followed by *Paenibacillus* sp. JS-4 (18.2%). Inoculation of wheat plants with *Paenibacillus* sp. JS-4 showed a greatly significant increase in shoot and root dry weights. Besides, *Paenibacillus* sp. BJ-4 and *Paenibacillus* sp. SZ-10 had higher dry weights of shoot and root as compared to the controls (Fig. 4B). The effects of these two bacteria on wheat seedlings were equal to the positive control with chemical nitrogen fertilizer. In Fig. 4C, cucumber seedlings

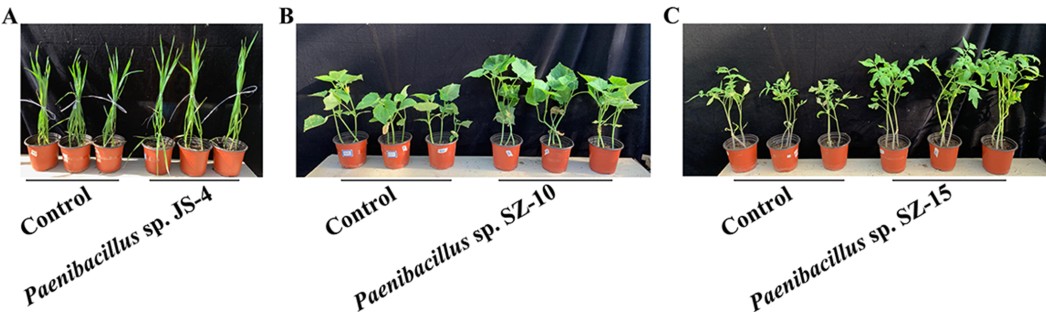

**Figure 3 Plant growth promotion by some *Paenibacillus* strains.** (A) Wheat seedlings inoculated with *Paenibacillus* sp. JS-4; (B) Cucumber seedlings inoculated with *Paenibacillus* sp. SZ-10; (C) Tomato seedlings inoculated with *Paenibacillus* sp. SZ-15.

inoculated with *Paenibacillus* sp. SZ-10 resulted in the highest heights both in shoot (50.0%) and in root (94.4%), followed by *Paenibacillus* sp. SZ-14 (33.7% and 38.7%, respectively) and *Paenibacillus* sp. WF-6 (18.4% and 62.4%, respectively). In addition, inoculation with *Paenibacillus* sp. SZ-10 presented the highest increase in dry weights of shoot and root of eight selected isolates, which showed more significant effect on cucumber seedlings than the positive control. Also, inoculation with *Paenibacillus* sp. SZ-14 had the second highest increase in total dry weight (Fig. 4D), which was the same as the positive control with chemical nitrogen fertilizer. Overall, *Paenibacillus* sp. SZ-10 showed significant growth-promoting effects on the cucumber plants. As shown in Figs. 4E and 4F, most isolates could promote growth of tomato. Out of them, inoculation with *Paenibacillus* sp. BJ-4 presented to enhance development of tomato length, both in shoot (64.6%) and in root (55.2%) (Fig. 4E). Inoculation with *Paenibacillus* sp. SZ-15 displayed maximum increases in shoot and root dry weights (Fig. 4F), which showed more promotive effect on shoot dry weight of tomato than the positive control.

## DISCUSSION

*Paenibacillus* species are ubiquitous in nature, and they are capable to form resistant endospores to allow them surviving in a wide range of environmental variables and to enhance plant growth by several mechanisms (*Bloemberg & Lugtenberg, 2001*). In this study, 179 bacterial strains were isolated by their growth on nitrogen-free medium from plant rhizospheres all over China. 16S rRNA sequence analysis showed that 25 of 179 bacteria belonged to *Paenibacillus* genus.

We revealed that 25 *Paenibacillus* strains had the *nifH* gene encoding the Fe protein of Mo-nitrogenase. Also, the 25 *Paenibacillus* strains exhibited nitrogenase activities. These results demonstrated that the 25 $N_2$-fixing *Paenibacillus* strains could provide nitrogen for plants. Phylogenetic analysis showed that the 25 $N_2$-fixing *Paenibacillus* strains were divided into five clusters. 20 of the 25 $N_2$-fixing *Paenibacillus* strains were in cluster I and cluster II that were closely related to *P. polymyxa*, *P. jamilae*, *P. peoriae*, and *P. brasilensis*.

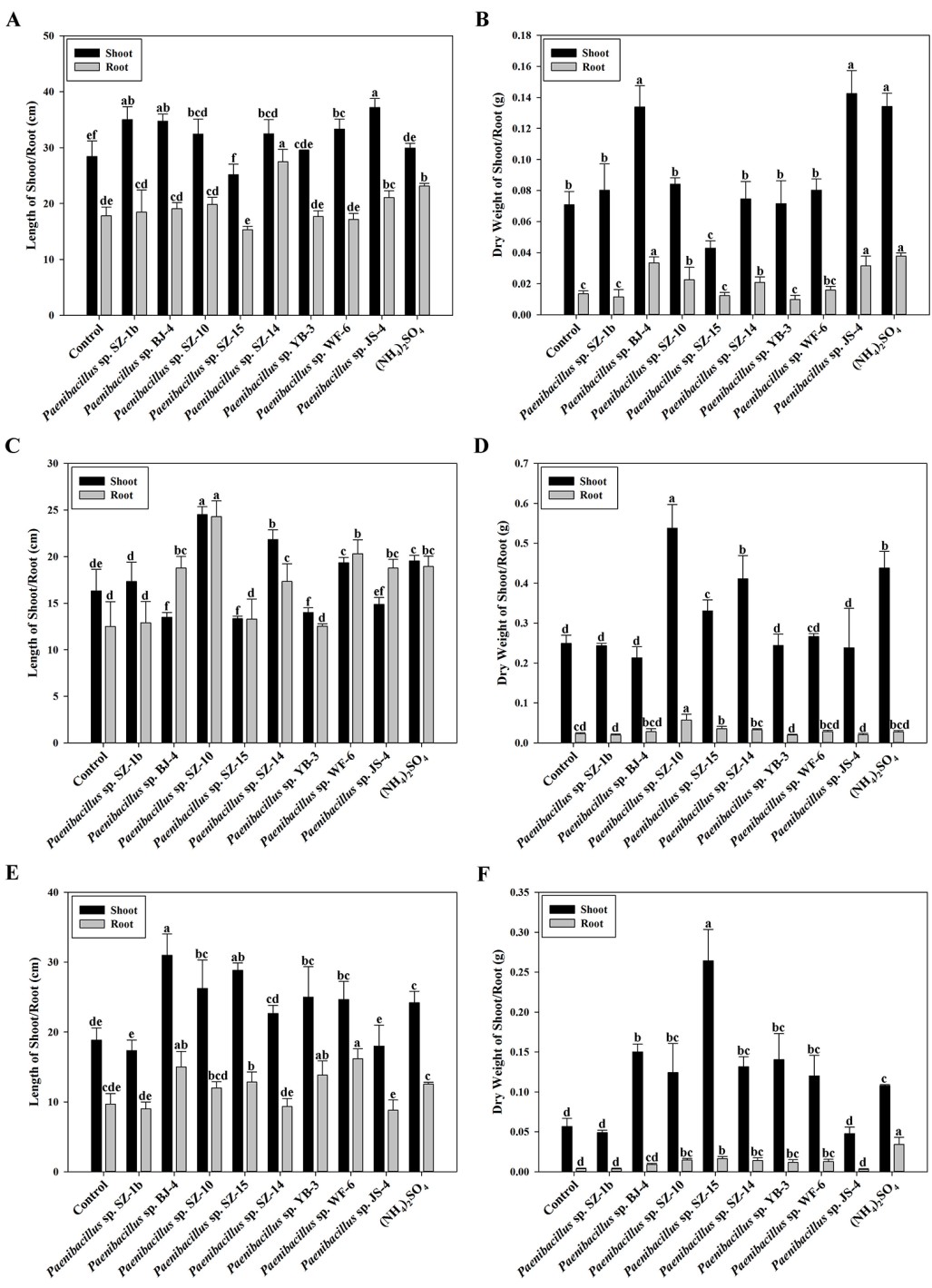

**Figure 4** **Effects of eight selected strains inoculation on shoot and root length of wheat (A), dry weight of wheat (B), on shoot and root length of cucumber (C), dry weight of cucumber (D), on shoot and root length of tomato (E), dry weight of tomato (F).** Control: un-inoculated seedlings. Date represent the means $\pm$ SE of 3 independent biological replicates. In the root group or shoot group, bearing different alphabets are significantly different from each other according to the LSD test ($p < 0.05$).

The other five $N_2$-fixing *Paenibacillus* strains belonged to cluster III, cluster IV and cluster V (including *P. jilunlii*, *P. zanthoxyli*, and *P. stellifer* mainly).

In this study, 20 of the 25 $N_2$-fixing *Paenibacillus* strains had inhibitory effects against plant pathogenic fungi, with average inhibition zones varying from 15 to 35 mm on plates. Especially, *Paenibacillus* sp. SZ-13b could suppress six tested bacterial plant pathogens. Wherease, *Paenibacillus* sp. SZ-1b, *Paenibacillus* sp. SZ-15, and *Paenibacillus* sp. JS-4 could suppress five tested bacterial plant pathogens with strong inhibition activities. The 20 strains with inhibitory effects against plant pathogenic fungi belonged to cluster I and cluster II that were closely related to *P. polymyxa*, *P. jamilae*, *P. peoriae*, and *P. brasilensis*. Our results are consistent with the previous results that *P. polymyxa* have long been known for their great ability to produce peptide antibiotics to suppress the growth of plant pathogenic fungi (*Deng et al., 2011*; *He et al., 2007*; *Helbig, 2001*; *Raza, Yang & Shen, 2008*). For examples, *P. polymyxa* M1 ( HE577054), which was isolated from root tissues of wheat, was able to promote wheat growth and suppress several phytopathogens (*Niu et al., 2011*; *Yao et al., 2008*). *P. polymyxa* SQR-21 (CP006872) selected from the rhizosphere soil of watermelon could significantly inhibit *F. oxysporum* (*Raza et al., 2009*). *P. brasilensis* PB1 72 (NR025106) isolated from the maize rhizosphere was able to protect seeds and roots against phytopathogenic fungi (*Fusarium moniliforme* and *Diplodia macrospora*) (*Von der Weid et al., 2005*; *Von der Weid et al., 2002*).

Additionally, 22 $N_2$-fixing *Paenibacillus* strains (except for *Paenibacillus* sp. AH-1, *Paenibacillus* sp. CD-4a and *Paenibacillus* sp. CD-4b) were capable of producing IAA, which is a primary plant hormone regulating plant growth and development. Among them, *Paenibacillus* sp. WF-6, *Paenibacillus* sp. SZ-1a, *Paenibacillus* sp. SZ-1b, *Paenibacillus* sp. BJ-5, *Paenibacillus* sp. YB-3 generated higher yield of IAA.

According to the results of nitrogenase activity, IAA level and inhibitory effect against plant pathogens, 8 strains were chosen to inoculate wheat seedlings, cucumber seedlings and tomato seedlings to analyse their plant promotion effects. We found that *Paenibacillus* sp. JS-4 and *Paenibacillus* sp. BJ-4 promoted wheat growth, as well as the chemical nitrogen fertilizer did. While *Paenibacillus* sp. SZ-10 and *Paenibacillus* sp. SZ-14 promoted cucumber growth as well as the chemical nitrogen fertilizer did. The two strains *Paenibacillus* sp. SZ-15 and *Paenibacillus* sp. BJ-4 significantly promoted tomato growth. Moreover, the 4 strains including *Paenibacillus* sp. SZ-10, *Paenibacillus* sp. SZ-14, *Paenibacillus* sp. YB-10, and *Paenibacillus* sp. WF-6 could promote tomato growth. From these results, we found that the plant promotion effects exhibited by a *Paenibacillus* strain varied among plants. At present, we do not know why a same *Paenibacillus* strain had different promotion effects on different plants.

Taken together, 25 $N_2$-fixning *Paenibacillus* strains were isolated from plant rhizospheres. The 5 strains including *Paenibacillus* sp. JS-4, *Paenibacillus* sp. SZ-10, *Paenibacillus* sp. SZ-14, *Paenibacillus* sp. BJ-4 and *Paenibacillus* sp. SZ-15 with the significant effects of promoting plant growth have great potential as bio-fertilizer.

Microbial fertilizers are widely used in plantation of vegetables in China. The members of *Bacillus* genus, such as *Bacillus subtilis*, *Bacillus amyloliquefaciens* and *Bacillus licheniformis*, are usually used in biofertilizers. The *Paenibacillus* strains with nitrogen fixation and

multiple bacterial properties for promoting plant growth obtained in this study have great potential to be developed as biofertilizers.

## CONCLUSION

In conclusion, 25 $N_2$-fixing *Paenibacillus* strains were isolated from plant rhizospheres. Most of them possessed multiple beneficial properties and characteristics of PGPR. They could fix atmospheric nitrogen, produce the profitable phytohormone IAA, control against a wide set of plant pathogens, and enhance growth of diverse important plants. In particular, the five strains including *Paenibacillus* sp. JS-4, *Paenibacillus* sp. SZ-10, *Paenibacillus* sp. SZ-14, *Paenibacillus* sp. BJ-4 and *Paenibacillus* sp. SZ-15 with the significant effects of promoting plant growth could be developed and commercially formulated to substitute for environmentally harmful chemical fertilizer and pesticides in field experiments.

### Funding

This work was supported by the National Key Research and Development Program of China (Grant No. 2017YFD0201705) and the National Nature Science Foundation of China (Grant No. 31770083). The funders had no role in study design, data collection and analysis, decision to publish, or preparation of the manuscript.

### Grant Disclosures

The following grant information was disclosed by the authors:
National Key Research and Development Program of China: 2017YFD0201705.
National Nature Science Foundation of China: 31770083.

### Competing Interests

The authors declare there are no competing interests.

### Author Contributions

- Xiaomeng Liu conceived and designed the experiments, performed the experiments, analyzed the data, prepared figures and/or tables, authored or reviewed drafts of the paper, approved the final draft.
- Qin Li contributed reagents/materials/analysis tools.
- Yongbin Li analyzed the data, contributed reagents/materials/analysis tools.
- Guohua Guan analyzed the data.
- Sanfeng Chen conceived and designed the experiments, analyzed the data, approved the final draft.

### Data Availability

The raw data are available in the Supplemental Files.

## Supplemental Information

Supplemental information for this article can be found online at http://dx.doi.org/10.7717/peerj.7445#supplemental-information.

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
