# Peer review of "Paenibacillus strains with nitrogen fixation and multiple beneficial properties for promoting plant growth"

_PeerJ, doi:10.7717/peerj.7445_

## Round 0.1 · original submission · Major Revisions

We have received three detailed reviews. Please address their comments, especially regarding language quality and text flow, the lack of comparative pot trials with nitrogen chemical fertilizer amendment and the detail present in the materials and methods section.

Reviewer 1 ·

Basic reporting

The English language should be improved throughout the manuscript to improve overall clarity. The current phrasing makes comprehension difficult at many places like line 52, 61-62, 76, 79, 144, 149, 151, 156-159, 172, 174, 212-214, 263 etc.)

Do not repeat the methodology or provide background information in the ‘Results’ section (for example, line 172-174, 227-232). Please revise it extensively.

List names of all bacterial species in Table 1 for better understanding and clarity.

Please see the comments on the annotated copy for further details.

Experimental design

Discuss the main research questions that you wanted to address in this study at the end of the Introduction section.

In my opinion, the Introduction section is too short and enough background and rationale behind this study are not provided.

Details about the sampling sites and plant type are missing in the Materials & Methods section. Plant-type has a direct influence on the rhizospheric bacterial population so brief information about the plants is crucial to provide. The authors can refer to Table 1 here.

Overall in the ‘Materials & Methods’ section authors should thoroughly revise and provide enough details to replicate the procedures. In its current state, the information is ambiguous. Please see the comments on the annotated copy for more details.

Line 194: It is not clear, how all strains can be identified as Paenibacillus? Were the authors specifically targeting Paenibacillus genus in this study? This again links back to the lack of research questions and hypotheses for this study.

Validity of the findings

Some of the information in the Results section is difficult to understand, partly because it cannot be linked to the questions that the authors were trying to answer in this study and partly due to lack of details in the Materials and Methods section or some random new information. Please see the comments on the annotated copy for more details.

Most of the information in the Discussion section is just repetition. Please proof-read it and thoroughly modify it.

Additional comments

Overall, this study highlights the various benefits that strains of Paenibacillus genus can provide to plants. There was significant amount of data generated in this study, however, many pieces of the information are missing in this manuscript in its current state, which reduces its scientific relevance.

Annotated reviews are not available for download in order to protect the identity of reviewers who chose to remain anonymous.

Reviewer 2 ·

Basic reporting

The Manuscript Number #35859: "Isolation, identification and characterization of Paenibacillus strains with potentials for biofertilizer" is a good research paper for promoting sustainable agriculture by using Paenibacillus strain as potential Nitrogen biofertilizer for vegetable production and an alternative of N2 fertilizer. Authors attempt to screen the nifH gene in isolated strains and also checked its nitrogenase activity along with antagonistic activity with the most common fungal pathogen for selection of efficient biofertilizer. However, the authors should carefully examine the whole manuscript and improve as per the following minor comments:

General comments:

1. Authors should include some recent references in the introduction part and also discuss the nifH gene mechanism in plant growth promotion section.
2. Authors should add some information regarding the recent scenario in nitrogen chemical fertilizer usage nationally and globally. And also, how it plays a role in environmental disturbance and loss of soil health.
3. The authors only tested the effect of biofertilizer on plant growth without any comparative pot trials with nitrogen chemical fertilizer amendment. How the authors justify this when the objective of the study is to provide efficient nitrogen biofertilizer under sustainable agriculture practices.
4. Authors should briefly explain how the nifH gene plays a role in nitrogen fixation in discussion section with nitrogenase activity in identified strains.
5. Authors should justify why they selected Wheat, Cucumber and Tomato as test plants during pot trials under this study.
6. Authors should include pot trials figures/images that showed the effect of biofertilizer.
7. If possible, author should give the GPS location of soil sample collection.
8. Authors should mention the significant difference on mean data value of Figure, 2, 3 and table 1 by different alphabet like. a, b, c,d, e………... so this more better for understanding between difference treatment combination.
9. Authors should mention the future challenges about microbial consortium or biofertilizers for agricultural production at farmers fields.
In last this can be accepted after minor revision.

Experimental design

ok.

Validity of the findings

good

Additional comments

The Manuscript Number #35859: "Isolation, identification and characterization of Paenibacillus strains with potentials for biofertilizer" is a good research paper for promoting sustainable agriculture by using Paenibacillus strain as potential Nitrogen biofertilizer for vegetable production and an alternative of N2 fertilizer. Authors attempt to screen the nifH gene in isolated strains and also checked its nitrogenase activity along with antagonistic activity with the most common fungal pathogen for selection of efficient biofertilizer. However, the authors should carefully examine the whole manuscript and improve as per the following minor comments:

General comments:

1. Authors should include some recent references in the introduction part and also discuss the nifH gene mechanism in plant growth promotion section.
2. Authors should add some information regarding the recent scenario in nitrogen chemical fertilizer usage nationally and globally. And also, how it plays a role in environmental disturbance and loss of soil health.
3. The authors only tested the effect of biofertilizer on plant growth without any comparative pot trials with nitrogen chemical fertilizer amendment. How the authors justify this when the objective of the study is to provide efficient nitrogen biofertilizer under sustainable agriculture practices.
4. Authors should briefly explain how the nifH gene plays a role in nitrogen fixation in discussion section with nitrogenase activity in identified strains.
5. Authors should justify why they selected Wheat, Cucumber and Tomato as test plants during pot trials under this study.
6. Authors should include pot trials figures/images that showed the effect of biofertilizer.
7. If possible, author should give the GPS location of soil sample collection.
8. Authors should mention the significant difference on mean data value of Figure, 2, 3 and table 1 by different alphabet like. a, b, c,d, e………... so this more better for understanding between difference treatment combination.
9. Authors should mention the future challenges about microbial consortium or biofertilizers for agricultural production at farmers fields.
In last this can be accepted after minor revision.

Reviewer 3 ·

Basic reporting

Reviewers comments to author
Authors should answer following questions in the paper
1. Line no 1, Authors can slightly modify the title of the paper “Isolation, identification and characterization of Paenibacillus strains with potentials for biofertilizer” for making it attractive.
2. Again in the abstract line no 29 plant hormones production induced systemic resistance or nutrient mobilization. Instead of or it should be induced systemic resistance and nutrient mobilization
3. line no 32. Again in the abstract “fourth-fifth of them” word is not correct. This can be replaced with percentage or any number of microbes. Sentence should be reframed.
4. in the abstract “In this study, twenty-five Paenibacillus bacteria were isolated from plants rhizosphere and identified based on their 16S rRNA”. Sentence is unclear should be reframed
5. Abstract have many grammatical errors, many sentences are unclear and need to be reframed.
6. In line no 40 word “provoked” should be replaced.
7. Line no 68 plant pathogens should be replaced by plant pathogens.
8. In line no 74 under the heading material and methods, Sample collection isolated, procedures and culture conditions needs to be reframed.
9. Line no 197 sentence is unclear
10. In line no 162 the word reaped should be replaced by harvested.
11. In line no 163 Lengths is not correct.
12. Paper has many unclear sentences, grammatical errors and spelling mistakes which need to be corrected

Experimental design

Yes It is under scope of the journal.
Yes research question is well defined. although its not entirely novel work.
Methods are sufficiently described.

Validity of the findings

Findings are expected by using the strain, most of the research are going in the same direction.
Data is good and sound.
Conclusions are well stated.

---

## Round 0.2 · Minor Revisions

There seems to be some mistakes in the labelling of the columns in Fig.4: for example, in panel A the letters depicting significant diffferences in "shoot" / "control" are "ef", and the ones in "root" / " Paenibacillus SZ-15" are "e" implying no significant differences between them even though "root" / " Paenibacillus SZ-15" is much lower than almost all other columns and its standard deviation is minuscule. Similar incongruences are seen in other graphs. Please check them carefully.

Reviewer 2 ·

Basic reporting

Authors has been revised and modified manuscript as per my comments. they have mentioned all red color text in revised manuscript. now this can be accepted for publication.

Experimental design

this is ok.

Validity of the findings

ok.

Additional comments

Authors has been revised and modified manuscript as per my comments. they have mentioned all red color text in revised manuscript. now this can be accepted for publication

---

## Round 0.3 · accepted · Accept

Thank you for addressing the issues that remained.